# Current State of Research on the Mechanism of Cavitation Effects in the Treatment of Liquid Petroleum Products—Review and Proposals for Further Research

Denis Kuimov *, Maxim Minkin, Alexandr Yurov and Alexandr Lukyanov

Don State Technical University, Rostov-on-Don 344003, Russia; msi_58@mail.ru (M.M.);
yurov_dstu@rambler.ru (A.Y.); alexlukjanov1998@gmail.com (A.L.)
* Correspondence: kuimov_d@list.ru

**Abstract:** Cavitation, as a unique technology for influencing liquid substances, has attracted much attention in the oil refining industry. The unique capabilities of cavitation impact can initiate the destruction of molecular compounds in the liquid. At the same time with a large number of successful experimental studies on the treatment of liquid hydrocarbon raw materials, cavitation has not been introduced in the oil refining industry. Often the impossibility of implementation is based on the lack of a unified methodology for assessing the intensity and threshold of cavitation creation. The lack of a unified methodology does not allow for predicting the intensity and threshold of cavitation generation in different fluids and cavitation-generating devices. In this review, the effect of cavitation on various rheological properties and fractional composition of liquid hydrocarbons is investigated in detail. The possibility of using the cavitation number as a single parameter for evaluating the intensity and threshold of cavitation generation is analyzed, and the limitations of its application are evaluated. The prospects of introducing the technology into the industry are discussed and a new vision of calculating the analog of cavitation numbers based on the analysis of the mutual influence of feedstock parameters and geometry of cavitators on each other is presented.

**Keywords:** cavitation; hydrodynamic; acoustic; oil; asphaltenes; paraffins; viscosity reduction; desulfurization; light fractions yield increase

## 1. Introduction

### 1.1. Cavitation

Cavitation, in simple terms, describes the process of formation in a liquid followed by the collapse of bubbles. However, cavitation is difficult to support and mathematically hard to describe processes, which we observe every day in our daily lives. Every morning, when heating water for tea, one sees cavitation in the form of collapsing bubbles, making the familiar noise. Sound impulses are one of many energetic effects of cavitation.

For a long time, however, cavitation was primarily studied as a negative influence. The discovery of cavitation was made by chance during the launching of the British Navy destroyer "Daring" in 1893. During the tests, the design speed of the destroyer could not be achieved, which turned out to be caused by the process of formation and collapse of cavitation bubbles [1], first described as propeller slippage. Further research continued with the efforts of Reynolds in 1886 and was accelerated by the work of Rayleigh in 1917 [2]. From this period onwards, the research on cavitation was carried out from the point of view of minimizing the negative effects. The high energy density created per unit volume of the matter when rotating propellers can cause some negative effects, including high noise (collapsing bubbles cause acoustic emissions, highly undesirable for naval vessels), erosion (destruction of propeller surfaces), and additional vibration. [3] Such close attention, long study of the process, and high energy effects could not but lead scientists to the path of useful application of cavitation. Many researchers discovered a great potential for

influencing liquid materials to change their physical and chemical properties. Discovering the potential of cavitation effects, researchers have applied them to the implementation of technological processes in various industries, including the food industry [4], wastewater treatment [5], biomedical applications [6], treatment of liquid hydrocarbons to change their rheological properties [7–9], etc.

Cavitation is described as a process of the formation of micro-bubbles of vapor in a liquid product to be treated. In general, cavitation is described as a process of the formation of micro-bubbles of vapor in a liquid product to be treated. In general cases, the cavitation process can be represented as a process of formation of cavitation caverns in a liquid medium, subsequent growth, and collapse which leads to the release of high energy density. The process can be represented as a process of formation of cavitation caverns in a liquid medium, subsequent growth, and collapse which leads to the release of high energy density. A general picture of cavitation bubble formation is as follows. In the rarefaction phase, a cavity-like void is formed in the liquid and filled with saturated liquid. A general picture of cavitation bubble formation is as follows. In the rarefaction phase, a cavity-like void is formed in the liquid and filled with saturated liquid vapor. In the contraction phase, the cavity collapses under the action of increased pressure and surface tension forces. A significant effect of the cavitation process is due to the high concentration of energy released during the collapse process in the medium to be treated. In the contraction phase, the cavity collapses under the action of increased pressure and surface tension forces. A significant effect of the cavitation process is due to the high concentration of energy released during the collapse process in the medium to be treated. At the moment of collapse, gas pressure and temperature reach significant values, and according to some reports, they reach 100 MPa and 1000 °C, respectively [1]. This phenomenon is explained by the small volume of the substance at the time when the bubble reaches its minimum radius before collapsing. Based on the results of scientific studies [10–12], it can be said that the radius of a cavitation bubble during collapse can usually reach 10-7-10-8 m against the radius in the equilibrium state of 10-10-6 m. Variations of the volume of the cavitation bubble as high as thousandths lead to high values of stored energy. Acoustic wave research has been studied in compressible fluids [13,14].

Each single cavitation bubble can locally create high energy density in some volume unit of the substance. While creating and maintaining stable cavitation in the treated liquid medium, millions of bubbles ideally throughout the volume of the substance create and collapse simultaneously, creating local areas of increased temperature and pressure throughout the volume of the substance, keeping overall parameters consistent with the environment, and thereby making it possible to initiate chemical reactions that would require large amounts of energy in normal conditions.

Cavitation is created by a change in pressure in the liquid, while the change in pressure and rupture of the liquid can be achieved in different ways. In [15], authors distinguished four main types of cavitation that have different physical mechanisms of creation:

1. The hydrodynamic cavitation;
2. Acoustic cavitation;
3. Optical cavitation;
4. Particle cavitation.

### 1.1.1. Acoustic Cavitation

Acoustic cavitation is the process of a liquid rupturing when acoustic vibrations (high amplitude ultrasonic signal) pass through it. When the liquid is brought into contact with ultrasound, the cavity within the liquid begins to vibrate in conjunction with the sound waves. When the cavity goes through a compression cycle, its internal pressure increases, while in the rarefaction phase, its local pressure decreases. The formation, subsequent growth in the rarefaction phase, and the implosive collapse of these cavities in the compression phase are the essence of acoustic cavitation.

Processing of raw materials by acoustic cavitation is more often used in technological processes to change the fractional state of the substance without significant heating. This effect has some advantages over hydrodynamic cavitation. During the operation of hydrodynamic cavitation energy transfer to the system is accompanied by a transformation of the mechanical energy of the moving parts (creating cavitation) into thermal energy. Thus, the priority way to intensify the phenomenon of cavitation, until recently, was the ultrasonic effect on the liquid. There are various types of ultrasonic reactors providing treatment of liquid by acoustic cavitation, differing by sources of ultrasonic oscillations (magnetostriction, piezoelectric).

Experimental studies of high-intensity ultrasound treatment of oil showed the possibility of initiating paraffin degradation processes, allowing to reduce of the viscosity of treated oil and petroleum products by 20–30% [16,17], simplifying and reducing the cost of transportation of raw materials in pipelines, and eliminating the possibility of viscous oil deposits on the pipe walls.

Yet, for all the positive results of the treatment, the development of effective devices is fraught with difficulties. Most devices that use acoustic cavitation energy cannot ensure even treatment of the fluid over the entire volume of the working chamber with a large cross-section. The design of such devices requires a careful selection of acoustic properties, as well as materials. This problem is particularly acute in the treatment of highly viscous liquids, such as oil and oil residues, characterized by a high degree of attenuation of ultrasonic waves [18]. All this leads to the necessity of reducing the flow velocity inside the working chamber to provide a complete treatment of the liquid through the entire volume of the raw material. Therefore, in practice, the use of acoustic influence is possible only to reduce the near-wall viscosity of oil during oil pumping through pipelines. Since oil is a non-Newtonian fluid, it is subject to the rule that the viscosity of the medium decreases with increasing applied stress—the medium slides along a solid smooth surface. Exposure to acoustic vibrations at a single point in a pipeline can instantly reduce the viscosity of a thin near-wall layer of oil. This effect can propagate for hundreds of meters, due to the reflection and high degree of propagation of acoustic waves through the pipeline. Thus, the effective use of acoustic oscillations is not possible to change the rheological properties of oil and oil residues of large volumes. As a result, the use of acoustic cavitation generators in the refining industry is not feasible.

### 1.1.2. Hydrodynamic Cavitation

Hydrodynamic cavitation is the process of the formation of cavities or bubbles in a liquid filled with vapor and gas under the local pressure reduction produced by the fluid flowing around solid bodies. The most common design of a hydrodynamic cavitation generator is a Venturi tube. Venturi tubes, for all their simplicity, have several different configurations (Figures 1 and 2), however, the common feature of the operating principle is the reduction of fluid pressure by placing constrictions in the path of the moving fluid. The tapering device, which is the main working organ of the apparatus, consists of a baffle with an orifice. The high-pressure fluid moves towards the baffle and, passing through the orifice is accelerated. An increase in velocity will result in an inevitable increase in kinetic energy and a drop in potential energy represented by pressure. When velocity and pressure reach certain thresholds for a given medium, there will be a violent release of dissolved gas in the liquid, creating cavitation.

The implement, a tapering device, can be made with one or more conical or cylindrical orifices (Figure 3).

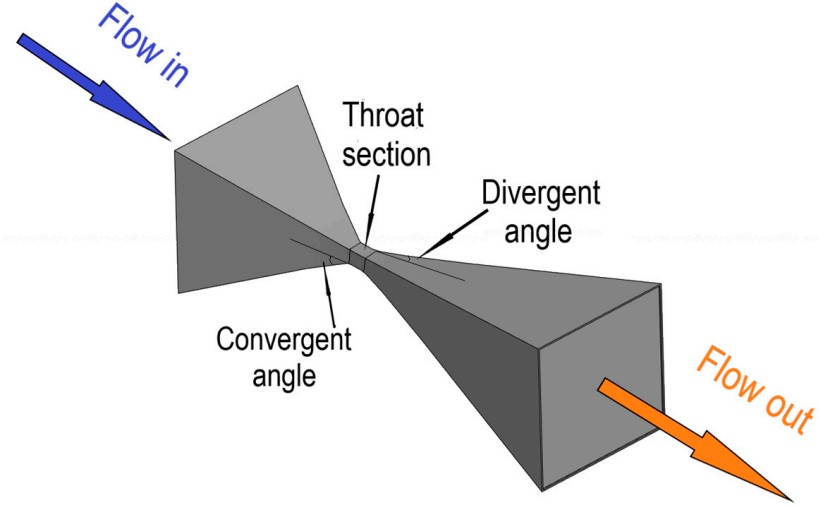

**Figure 1.** Venturi slotted tube configuration [19].

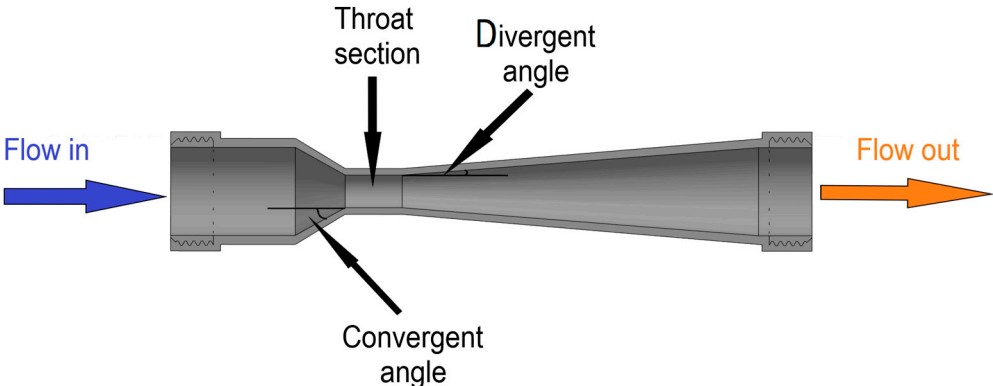

**Figure 2.** Venturi ring tube configurations [19].

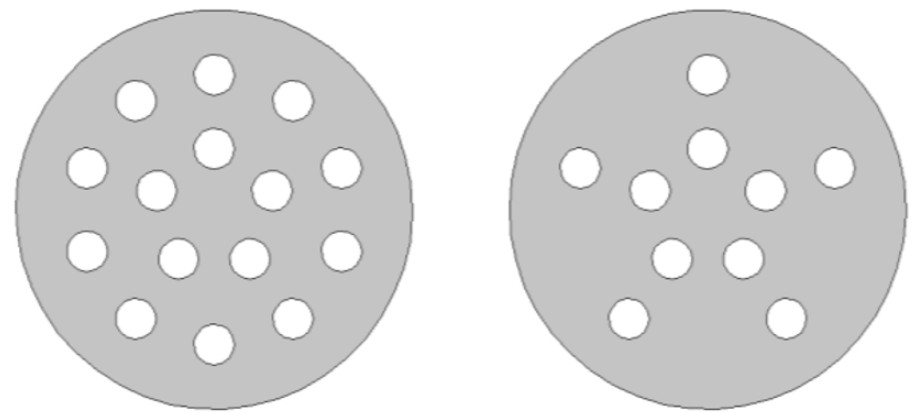

**Figure 3.** Diaphragm plate configurations.

Vortex devices consist of three main parts, including a cavitation bubble formation chamber, a tangential tube, and an axial tube (Figure 4). According to the operating principle [20], the flow of the treated liquid feedstock enters the chambers through the tangential tube, overcomes the pressure differential, and forms a vortex flow in the chamber This occurs when the local pressure drops below the water vapor pressure. The cavitation cavities grow as they move toward the outlet. On reaching the axial tube, the pressure is restored, and the cavity collapses by implosion.

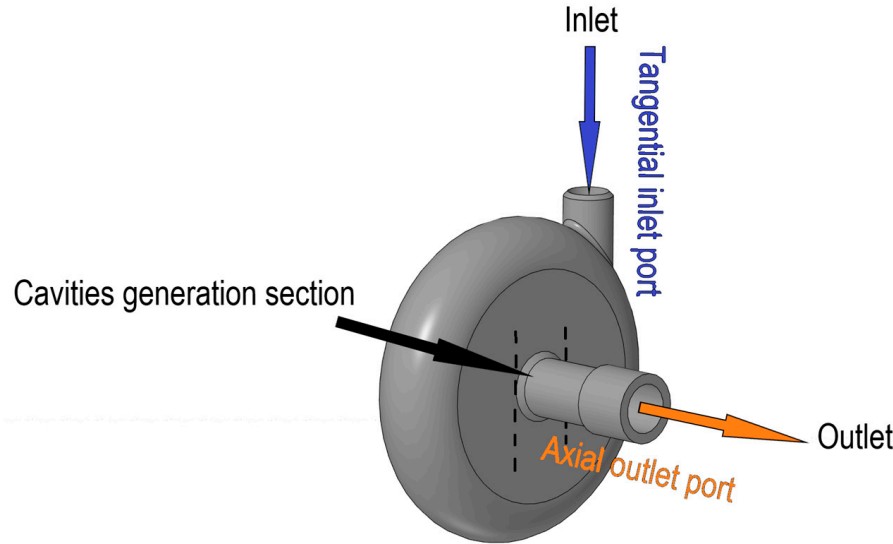

**Figure 4.** Vortex apparatus configurations [20].

Devices with discrete secondary parts as hydrodynamic cavitation generators are relatively new. Such devices consist of four main elements: inductor, winding installed in the inductor grooves, the working chamber of a cylindrical shape, made of nonmagnetic material and located inside the inductor, a set of ferromagnetic elements, located in the working chamber (Figure 5). The principle of the operation of such devices consists of the movement of a large set of ferromagnetic elements throughout the volume of the working chamber, under the influence of the external electromagnetic field generated by the inductor with the winding.

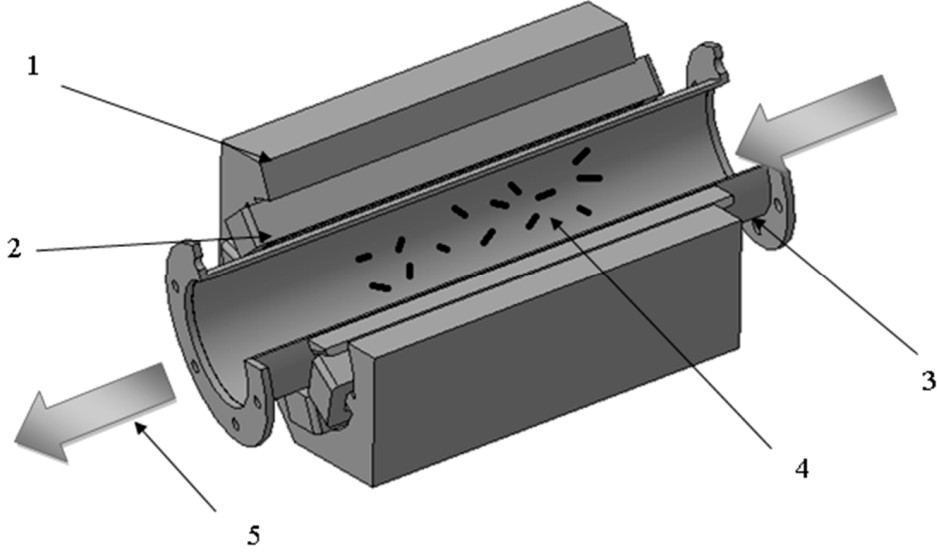

**Figure 5.** Device with discrete secondary part: (1) device body; (2) concentrated winding; (3) tube of non-magnetic material; (4) ferromagnetic elements; (5) flow of treated material.

For a long time, devices with a discrete secondary part were studied as vortex layer generators. The basic energy effect was the collision energy of the elements with the walls of the cavity and with each other. The intense movement of ferromagnetic elements through the liquid can also create hydrodynamic cavitation. Certain changes in the design [21,22] can increase hydrodynamic cavitation and decrease the number of collisions between the elements.

### 1.1.3. Optical and Particle Cavitation

Optical cavitation and particle cavitation are not used on an industrial scale but are used mainly for experimental studies to study the shapes of cavitation bubbles. Optical cavitation can be created either by short laser pulses focused on a solution with a low absorption coefficient or by using a continuous laser [23–25]. Cavitation created by continuous lasers in the dye-doped binary solutions was first described by Rastopopov and Sukhodolsky back in 1990 [23,24]. The mechanism of thermal cavitation in this case is the creation of a superheated area at the focal point, initiating intense evaporation of the liquid. Created bubbles falling into colder region collapse, releasing accumulated energy. Meanwhile, interest in thermal cavitation is rapidly growing due to the spread of optical cavitation technology, created by short laser pulses. Particle cavitation is created by a beam of elementary particles, such as a neutron beam that ruptures a liquid, as in the case of the bubble chamber. Interest in these types of cavitation is determined by the possibility of the creation of cavitation cavities of the required shape in the required space area, allowing the carrying out of fundamental investigations in the laboratory.

### 1.2. The Effects of Cavitation

Cavitation describes the formation of the vapor phase in a liquid when it is subjected to a reduced pressure at a constant ambient temperature [26]. When the vapor pressure of the liquid drops below the pressure of the liquid, small bubbles (cavities) form inside the processed liquid medium. When the formed bubbles reach a higher pressure area, their collapse occurs, which is called cavitation. The collapse of the bubbles may cause pressure peaks of up to 100 MPa [27] and if the bubble is compressed asymmetrically, so-called micro-jets may be formed with high velocities exceeding 100 m/s [28]. In addition, so-called hot spots with extreme temperatures of the order of several 1000 K may form in the center of the bubble when it collapses [29], which may cause the formation of highly reactive radicals [30]. Both liquid qualities (temperature, density, viscosity, and surface tension) and the presence of impurities (amount of solids and amount of dissolved gases that can act as nuclei) influence the manifestation of cavitation.

Liquid vapors, dissolved gases, and substances with high vapor pressure can enter the cavity while molecules and ions of dissolved non-volatile substances cannot [31]. The energy released by the collapsing cavitation bubbles is sufficient to excite, ionize, and dissociate water molecules, and gases inside the cavitation cavern [16]. Physical effects include turbulence, shock waves, mixing, and erosion. Chemical effects include sonoluminescence and radical formation (OH* and H* for water). The collapse of cavitation bubbles in most cases is an asymmetric collapse resulting in shock waves and micro-jets [32], which are typical manifestations of cavitation. When a cavitation bubble comes into contact with a solid surface during compression, it can generate micro-jets that can impact the material surface at high velocity, thereby helping to clean the solid object or causing its collapse [33]. In [34], researchers demonstrated the results of an acoustic cavitation-generated bubble collapse study, according to which the shock wave velocity was 1500–2520 m/s and the pressure was 1.3 GPa.

### 1.3. Cavitation Mechanism

Cavitation Parameters

In the scientific literature, the cavitation number is often used as the only parameter for evaluating the intensity and threshold of cavitation creation. At the same time, a review of the scientific literature devoted to experimental studies of the impact of cavitation on the treated feedstock revealed contradictions in the publications. Analyzing the publications, one can come across the fact that during the processing of different liquids, cavitation numbers can be in a wide range of values. At that, a cavitation number can be created both at a cavitation number value equal to 0.5 and at a cavitation number greater than 10. Thus, it is no less important to determine the cavitation number, the parameter that is currently

the main one for estimating the cavitation rate in the reactor under study. Cavitation is divided into four main modes:

1. Non-cavitation flow;
2. Limited cavitation;
3. Evolving cavitation;
4. Supercavitation.

These postulates stand for quite a long time and are used by many scientists for preliminary estimation of the efficiency of possible generation of cavitation in a multicomponent liquid mixture. However, the results of scientific studies in recent years have shown that the dimensionless cavitation number cannot be widely used as the only parameter to estimate the probability of the creation of a stable cavitation cloud in the treated medium. In [35], scientists investigated the effect of hydrodynamic cavitation (Venturi tube) on dye. According to the results received by scientists, hydrodynamic cavitation is observed at cavitation number values from 0.13 to 0.18. In [36] at the decomposition of aqueous KI solution by hydrodynamic cavitation, the optimum cavitation number found by scientists was in the range from 0.15 to 0.25. In [37], the optimum cavitation number for the realization of the insecticide degradation process was taken as 0.067. However, in [38], the hydrodynamic cavitation treatment becomes possible at a cavitation number of 0.4. In [39], the cavitation number is chosen in the range of 0.1–1, and in [40] it is stated that cavitation can occur only at a cavitation number equal to 1.

The second problem, however, is not the problem of using the cavitation number as the only parameter of cavitation intensity but the way of determining the cavitation number. For stationary systems, which are Venturi tube and diaphragm plate tube, the most popular expression is the following:

$$c = \frac{p_{in} - p_s}{0.5\rho V^2} \tag{1}$$

where $P_2$ and $P_s$ are inlet and saturation pressures, $\rho$ is the density of the liquid to be treated, and $V$ is the velocity in the throat.

At the same time, along with common expressions for the definition of cavitation number, alternative ways of calculating this parameter have been developed, including different combinations of parameters of the pressure of liquid feedstock at the inlet and outlet of investigated stationary cavitation reactor:

$$c = \frac{p_{in} - p_s}{p_{in} - p_{out}} \tag{2}$$

$$c = \frac{p_{out} - p_s}{p_{in} - p_{out}} \tag{3}$$

$$c = \frac{p_{in} - p_{out}}{p_{out} - p_s} \tag{4}$$

$$c = \frac{p_{in} - p_{out}}{0.5\rho V^2} \tag{5}$$

where $P_{in}$ and $P_{out}$ are inlet and outlet pressures of cavitation reactors, respectively.

A large number of combinations of the same parameters in the expression of the cavitation number only confirms the complexity and probably inexpediency of using the presented parameter as the only parameter of cavitation intensity. A striking example is the work [41], where the authors evaluated the influence of the Venturi tube geometry on the cavitation number. The result of this study was a range of dimensionless cavitation numbers varying from 1.2 to 168, which showed how strongly the cavitation number depends on the location of the working chamber of the device where measurements are made. Another explanation for the large number of expressions for the definition of the cavitation number, and inconsistencies in the obtained values is the lack of a unified standardization in the

development and design of the hydrodynamic cavitation device. Since the technology of cavitation action has not yet advanced beyond laboratory studies, each scientific team carries out the design of the installation based on its own vision of the effectiveness of a particular design. Each laboratory setup of each individual team is unique. Moreover, the application of the developed setup is carried out for a particular liquid raw material whose parameters and characteristics are often unknown to many scientists.

In work [42], an estimated comparison of the numerical prediction of cavitation numbers with the results of experimental investigations was carried out. Numerical analysis of cavitation was performed using the most common model of Reynolds averaged Navier-Stokes equations in the Ansys Fluent software package. The main drawback of existing expressions for determining the cavitation number is the lack of consideration of the reactor working chamber geometry, which was demonstrated in this work [42], where there was no pattern of change in the cavitation number when changing the geometry of the tubes under study. At the same time, cavitation was observed both at a cavitation number equal to 1 and at a cavitation number equal to 5. It was also shown in this work that the temperature dependence of cavitation is different for each tube geometry. In general, the conclusions of the experimental study [42] only confirm the impossibility of using the cavitation number as the only parameter for predicting the occurrence of cavitation.

A difficulty in the development of an expression reflecting the cavitation number is also introduced by the extension to replace stationary cavitating reactors of installations with moving internal working parts, including vortex reactors, rotor installations, and devices with discrete secondary parts, for which the previously developed expressions are not applicable.

### 1.4. The Need for a Review

To date, a large number of studies have been conducted and many experimental results have been published in the scientific literature on the study of individual parameters of liquid feedstock on the intensity, aggressiveness, and threshold of cavitation creation. The influence of dissolved gases, solid impurities, the concentration of dissolved salts, density, and viscosity of raw materials on the intensity and threshold of cavitation creation have been studied separately. In most studies, scientists artificially create controlled conditions by adding certain substances to evaluate their effect on the creation threshold and intensity of cavitation. Controlled conditions are also created by scientists during experimental research on the treatment of various complex multicomponent liquids in order to confirm the effectiveness of the cavitation effect [43,44]. To date, in the scientific literature, there are a large number of studies confirming in laboratory conditions the effectiveness of the application of cavitation for the treatment of wastewater, food products, and liquid hydrocarbons.

The technology of treatment of liquid petroleum products by cavitation is particularly popular among scientists [7,45]. The interest of scientists is justified not only by the prospects of the technology but also by the increase in the share of heavy oil. Crude oil has been and will continue to be for the next few decades, humanity's main energy resource, contributing to industrial growth. Crude oil is a multi-dispersed substance with a complex fractional composition, varying from field to field. At its most basic, crude oil can be divided into those with a viscosity of greater than 100 MPa or a relative density of greater than 0.934 $g/cm^3$ and those with a density of less than 0.87 $g/cm^3$. Oil can also be classified using API as light (31.3–390), medium (22.3–31.10), heavy (10–22.30), and extra heavy (<100) [46]. Heavy oil resources are quite extensive, with known and proven reserves of resources reaching at least 70 percent of total reserves [47].

The large reserves of heavy oil are partly preserved by the high production of light grades. Heavy oil is difficult to produce, difficult to refine, and difficult to transport because of its high viscosity and high content of harmful impurities in crude oil, including asphaltenes (10%) and sulfur (over 1.3%). At the same time, in the current conditions of the depletion of deposits of "light" grades of oil and the continued global demand for

oil and petroleum products in the coming decades, heavy oil is essential for the global industry, which confirms not only the search for new processing technologies but also the technology of oil production [48,49]. Thus, the need to use heavy oil requires the search for new refining methods, different from the classical ones, which facilitate the technological process, reduce viscosity to increase pipeline transportation, remove harmful impurities, and increase the yield of "light" fractions.

The development of new refining technologies can reduce environmental pollution in refining processes and increase interest in refined products. Cavitation, due to its ability to create a high energy density per unit volume of matter, can not only improve existing oil refining processes but also replace some of them. At the same time, for example, at least 5000 articles have been published in recent years, including those in highly ranked and cited journals, where the high efficiency of cavitation application in oil and petroleum products quality improvement technologies was noted. These results are very significant for the oil industry, as they can reduce the financial burden on refineries. Yet, with high results, huge potential, and rather inexpensive technology, why it has not reached production?

This is the most important issue around which this review article is built. The complexity of cavitation may lie in the variability of the parameters of the treated raw material during the implementation of the proposed technological process. The collapse of cavitation bubbles, as mentioned above, leads to the release of energy and a local increase in temperature and pressure. The process of collapsing cavitation bubbles not only increases the temperature of raw materials and, accordingly, changes the viscosity of the liquid but also changes the fractional composition of the liquid medium. For certain types of liquid materials, such as liquid hydrocarbons, a change in the fractional composition leads to a change in the viscosity and density of the medium.

Thus, to study the regularities of the effect of various parameters of the cavitation effect on the technological processes of treatment of polydisperse media and assess the feasibility or cancel the limitations of the dimensionless amount of cavitation on the intensity and threshold of creation of cavitation in a polydisperse liquid is necessary to study the effect of cavitation directly on the characteristics of the processed raw materials.

The results of this review can provide basic insights into the mechanisms of cavitation, the energetic effects created, and most importantly, the problems that hinder the implementation of the technology. The main focus of this review will be on the limitations of the dimensionless cavitation number in the design of industrial-scale plants and process systems for processing complex multi-component systems.

## 2. Materials and Methods

Every year in the scientific literature appear more and more new studies confirming in laboratory studies the effectiveness of application of cavitation in the treatment of liquid hydrocarbons. At the same time, there are practically no studies reflecting the results of introduction of the technology into industry. This fact is partly explained by the lack of a full understanding of the mechanisms of cavitation impact on the fractional composition of oil, as well as the lack of a unified methodology of application of cavitation and impossibility of predicting the intensity and threshold of cavitation in different liquids.

The attempt to characterize cavitation by a single parameter showing both the probability of creation of cavitation and its intensity is an important step towards further research towards the introduction of devices directly into industrial areas and requires a detailed theoretical study to find the main fluid parameters that affect cavitation, including its creation and maintenance.

However, cavitation, whether acoustic or hydrodynamic, is a complex physical phenomenon, the intensity and efficiency of which in the framework of the implemented technological processes, is highly dependent not only on the physical parameters and characteristics of the initial liquid raw materials but also on the changing characteristics of the liquid raw materials in the processing. Real liquids, unlike controlled liquid raw

materials in laboratory conditions, which are supposed to be processed on industrial scale, are often complex multi-dispersed systems.

The purpose of this study is to assess the limitations to the implementation of cavitation technology in industry. The cavitation properties of fluids are important in the design of any cavitation generators and the development of a new single parameter for assessing the intensity of cavitation. This paper detailed theoretical study of both the effect of cavitation on the treatment of multi-dispersed liquid and evaluation and recommendations for the development of new expressions of the dimensionless cavitation number.

Liquid hydrocarbons were chosen as an object of study of the cavitation mechanism on the properties of liquid products. Due to their complex fractional composition of liquid hydrocarbons, the impact of cavitation on various parameters of raw materials during treatment is more evident. In the presented work, some scientific publications connected with research on influence of cavitation on separate parameters of hydrocarbons within realization of the basic technological processes, including decrease in viscosity of oil, increase in a yield of light fractions, and desulfurization of oil are selected.

## 3. Results

### 3.1. Reduction of Oil Viscosity

The main effect of ultrasonic cavitation on crude oil, and at the same time its main disadvantage, is the short-term reduction in product viscosity. The inconstancy of the conservation of changes in rheological properties was noted by researchers [50] in 2000. In 2004, scientists [51] investigated the viscosity of the treated product during the first few seconds after treatment in a simple way, using a pipette and flow time calculation. The authors used the viscosity coefficient as the evaluation parameter. In all iterations of the test, a consistent effect was observed, in that the viscosity of the raw material decreased during the first 60 s of treatment and then began to increase again, and in some cases even above the initial values, when there was further treatment with ultrasound. The hypothesis of reorientation of the molecular compounds was given as an explanation by the authors.

All the scientific studies carried out in the field of cavitation effects on oil and petroleum products in one way or another confirm the possibility of viscosity reduction. However, most of the studies are not directed at the mechanisms of oil viscosity reduction by cavitation, but on the prospects of application of some reagents together with cavitation to reduce the viscosity. According to the reviewed scientific literature, the main reagent that accelerates viscosity reduction is a hydrogen donor. Hydrogen donors can be hydrogen itself [52], methane, formic acid, cycloalkylaromatic compounds [53,54], tetrahydronaphthalene [55,56]. The authors of the studies explain the mechanism of action of the hydrogen donor by the appearance of active hydrogen in the treated product, which can stop the activity of macromolecular free radicals.

In [57], the oil viscosity was reduced by 5.35% after 10 treatments with a cavitation jet. The addition of a hydrogen donor (tetrahydronaphthalene, THN) significantly accelerated the viscosity reduction. With the addition of 0.03 mass fraction of THN, the decrease in viscosity over 10 treatments increased to 11.03%, and an increase in the mass fraction to 0.03 mass fraction accelerated the decrease to 15.7%. A cavitation nozzle with a diffusion section was used. In this case, the authors noticed that the addition of THN significantly reduces the initial viscosity of the oil, so the viscosity reduction was not calculated from the initial state, but from the new viscosity value obtained after adding THN, but before treatment with cavitation. While the results were successful, it is worth noting that the authors did not pay enough attention to the viscosity change. It is known that viscosity influences the threshold of cavitation and its intensity. When changing the initial viscosity of the product, the parameters and characteristics of the system remained the same, which could have a significant impact on the intensity of the cavitation created. For completeness of the experiment in such studies, it was advisable to treat oil diluted with compounds that are not hydrogen donors and to compare the results obtained by treating oil diluted with hydrogen donors and other neutral compounds.

In [58], the authors studied the effect of hydrodynamic cavitation on crude oil from three fields, including paraffinic oils Shikhbagi, Bulla Deniz, and Shirvan Azerbaijan oil. Vortex heat generators (VTH) were used as cavitation devices. Based on their experimental studies, the authors concluded that paraffinic oil was more amenable to treatment although paraffinic oils do contain resins. The greatest effect was achieved when hydrodynamic cavitation and toluene or butyl acetate were used simultaneously as reagents. In the most successful experiment, viscosity reduction using hydrodynamic cavitation and the introduction of butyl acetate to paraffinic oil was as high as 42%. The use of hydrodynamic cavitation allowed an average of 17–19% of additional viscosity reduction for the studied oils.

The effectiveness of cavitation was also confirmed by other works [59,60].

### 3.1.1. Mechanism of Viscosity Reduction

Many studies on the effects of cavitation on liquid hydrocarbons are carried out on high-density hydrocarbon mixtures (above 50 API). It is known from the scientific literature that acoustic cavitation treatment of high-density hydrocarbons at 20 °C [61] is only related to temperature changes. For the technological process, viscosity reduction is more useful not temporarily related to temperature changes in the feedstock but having a long-lasting effect and related to the impact on the fractional composition. Based on this statement, the main mechanism to reduce oil viscosity should be associated with an impact on asphaltene, paraffin, and alkane molecules. It is the high content of asphaltene, paraffin, and alkane molecules that leads to an increase in oil density and viscosity and leads to changes in colloidal structure and rheological characteristics [62].

Although the main effect of cavitation on viscosity reduction is related to the effect of increasing feedstock temperature, an estimate of the residual viscosity reduction is presented in [62]. It is known that after some time after the treatment of hydrocarbon raw material, the viscosity of the treated product returns to almost its original state. In [63], it was found that after acoustic cavitation treatment for 8 min (46 kHz and 50 W output power), the permanent viscosity decrease was 10% as compared to the original product viscosity. This conclusion suggests that cavitation can not only increase the overall temperature of the feedstock but also affect the individual components of the liquid hydrocarbon product. The authors of [64] presented the results of studies, according to which treatment of crude oil by ultrasonic cavitation led to a decrease in the size of asphaltene clusters and a decrease in the total amount of asphaltenes, while the application of toluene solution increased the solubility of asphaltenes.

Thus, the mechanism of viscosity reduction is related not only to an increase in the temperature of the feedstock but also to the degradation processes of certain molecular compounds. In [65], the authors found that the viscosity of heavy grades of oil increases exponentially with asphaltene content; moreover, the viscosity of heavy oil samples decreases significantly when the temperature increases from 25 to 85 °C with a constant volume fraction of asphaltenes.

A number of authors, in accordance with the theory about the possibility of destruction of molecular compounds by cavitation impact, have presented results on the development of methods for calculating the destruction of molecular bonds. In work [66], the formula of molecular bond destruction theory is presented.

$$\tau_p = \tau_0 \exp\left\{ \frac{E_0 - \gamma(\sigma_c + \sigma_u)}{kT} \right\} \tag{6}$$

The presented equation shows the mechanism of bond failure where the process depends on the material temperature T, bond breaking energy $E_0$, static stress $\sigma c$ and ultrasonic stress $\sigma_u$. Γ characterizes the degree of average stress transfer per bond and depends on the structure, and k is Boltzmann's constant.

$$\sigma_u = \frac{1}{\tau} \int_0^{\tau_p} dt \sqrt{\sigma^2(t)} \equiv \int_0^{\tau_p} dt |\sigma(t)| \tag{7}$$

These equations allow at the stage of technological process development and analysis of future device efficiency to determine the feasibility of selected parameters and assess the probability of product processing.

In [67], the equations for the calculation of energy required for the destruction of molecular compounds and energy created by cavitation are presented. The bond-breaking energy of a chemical compound in one molecule is determined by the relation [67,68]:

$$E_p = \frac{E_c}{N_A} \tag{8}$$

The energy generated when the cavitation bubble collapse [67,69]:

$$E_k = 4\pi \left( R_0^3 - R_{min}^3 \right) \frac{P}{3} \tag{9}$$

The number of molecules with broken bonds at the collapse of one bubble is determined by the following expression:

$$N = \frac{E_k}{E_p} \tag{10}$$

Energy absorbed by the formation of cavitation bubble of radius $R_0$, filled with vapor, is calculated by the formula:

$$E_0 = 4\pi R_0^2 \sigma + 4\pi R_0^3 \frac{(P_0 + P_\text{п})}{3} \tag{11}$$

where the first term is the energy spent on the formation of the free surface of the bubble, the second term is the energy spent on the formation of the bubble cavity and the energy of filling this cavity with gas.

### 3.1.2. Reduction of Paraffin Content

Paraffin (C11–C20, boiling between 200 to 350 °C, C20–C35, boiling between 350 to 550 °C) forms crystalline lattices in the oil stream as the temperature decreases [70]. The strength of the paraffin structure increases as the temperature of the paraffin oil decreases, as more and more paraffin changes from liquid to solid state. The results of the review show that the effects of cavitation can damage the crystal structure of paraffin in crude oil, resulting in a decrease in viscosity.

In [71], a comparison was made between the acoustic cavitation treatment of crude oil from the Tahe field and cavitation with the addition of nickel oleate to the crude. Compared to the crude oil, the acoustic cavitation treatment increased resins, asphaltenes, and aromatic hydrocarbons and a decrease in saturated hydrocarbons authors suggest that the increase in resin content is due to the network structure of the resins dissolving some of the saturated hydrocarbons that are released by cavitation. The cavitation effect also dissipates the crystalline structure of the paraffin in the crude oil, resulting in lower viscosity. Under the influence of acoustic cavitation, paraffin with a long chain above C22 break down into alkanes with short carbon chains C12–C22 and less than C12.

### 3.1.3. Asphaltene Content Reduction

Asphaltenes are the heaviest fraction of crude oil and consist of molecules soluble in aromatic compounds and insoluble in paraffin [72]. The molecular weight of asphaltenes is difficult to estimate because of their self-aggregation and coagulation [7]. In [71], together with a study of the paraffin reduction process, studies have been carried out to find the mechanism of the effect of cavitation on asphaltenes. From the results presented in the paper, it can be concluded that for asphaltenes, there is a gradual increase in aromatic hydrogen index (Iar) and aliphatic side chain index (CH/CH2) due to the cavitation effect, which is consistent with the marked changes in aromatic hydrogen in asphaltenes. From the quantitative analysis of the saturated hydrocarbon gas chromatogram. It is shown that long-chain alkanes above C22 are cleaved to relatively short alkanes, and shorter alkanes can

be obtained with the help of nickel oleate. Finally, it is noted that the asphaltene structure was changed due to the cavitation effect, resulting in lower viscosity and improved crude quality. In addition, the addition of nickel oleate can enhance the viscosity reduction effect.

In [72], the authors proposed three main steps to reduce the viscosity of crude oil under the influence of acoustic cavitation. The first stage is the dissolution of suspended particles in crude oil, leading to an increase in viscosity. In the second stage, as the temperature of the crude increases, long chains and heavy molecules decompose, and free radicals are formed, all of which lead to a decrease in viscosity. At the final stage, the viscosity increases, which is explained by reaching a certain minimum of short-chain molecules and free radicals that start to reintegrate with each other. Thus, the authors made the key conclusion that during the acoustic cavitation treatment, there are always two active mechanisms: decomposition of asphaltenes to a certain minimum, followed by the formation of new asphaltenes of similar structure. At the same time, according to the given results, the final index of kinematic viscosity was higher than that of the original oil.

The authors [73] carried out viscosity reduction in vacuum residue from Hindustan Petroleum Corporation Limited, Mumbai. The asphaltene content was reduced from 13.5 wt.% to 8.1 wt.% after 90 min of acoustic cavitation treatment. Thus, when ultrasonic cavitation was applied in the absence of hydrogen donors and catalysts, a reduction of about 40% in asphaltene content in the residue was observed.

### 3.1.4. Viscosity and Cavitation

From the overview presented above, a small conclusion could be made about a large number of possible combinations of solutions for the implementation of the technological process of oil product viscosity reduction by cavitation. The presented results demonstrate not only the effectiveness of both acoustic cavitation and hydrodynamic cavitation but also a large number of possible reagents that accelerate the viscosity reduction process.

However, it is important to note that the high variability of the technological process creates a multitude of problems:

1. In each study, different combinations of feedstock and reagent are used. Each reagent has a different effect on the fractional composition of the oil. Since published papers mainly present results of the combined effect of cavitation and reagent on oil, it is difficult to evaluate the efficiency of reagent and cavitation separately.
2. The use of different grades of oil can have a significant impact on the final result. Both the fractional composition of the oil and the initial rheological characteristics of the feedstock may have an influence. For example, the article [58] states that paraffinic oil was better affected by cavitation. Organic impurities are an integral part of petroleum products. In general, oil consists of carbon, hydrogen, sulfur, oxygen, and nitrogen. In small doses, the oil may contain various metals, mostly nickel and vanadium, as well as a large number of impurities and gas inclusions that affect the threshold of cavitation creation and its intensity. The majority of authors do not take into consideration the composition of the hydrocarbon raw materials under treatment. They often ignore several impurities and other chemical compounds and evaluate only the target fractions. This approach does not allow estimating and comparing results obtained by different teams of scientists because both distorted cavitation characteristics due to ignorance of impurities and gas inclusions and oil fraction composition could have influenced the final result.
3. Adding a reagent reduces the initial viscosity of the product. Even if the viscosity of a mixture of reagent and oil is taken for subsequent experimental studies, this will also distort the final results. A change in rheological qualities while maintaining the characteristics of the cavitation generator affects the threshold for the creation of cavitation and its intensity.

Regarding the final point, both early studies and publications of the current period seldom consider the mutual influence of viscosity and cavitation. Changes in viscosity in one direction or another can have a direct impact on the dynamics of cavitation bubbles.

In [74], a numerical study of viscosity and its effect on microbubble formation was carried out using the finite volume method. The results of the study demonstrated that micro-jets cannot impact certain Reynolds numbers. In [75], the authors confirmed using the finite element method that viscosity can suppress the development of microjets in a liquid. Thus, the cavitation treatment process is dynamic, and the characteristics and degree of impact of cavitation on the liquid petroleum product will change in conjunction with changes in the qualities of the feedstock.

The authors [76] performed an experimental study of the effect of viscosity on the dynamics of cavitation bubbles. For the study, the authors selected five types of liquids (1000 mL deionized water (viscosity 0.0113 St), 200 mL glycerol + 800 mL deionized water mixture (viscosity 0.0314 St), 400 mL glycerol + 600 mL deionized water mixture (viscosity 0.0595 St), 600 mL glycerol + 400 mL deionized water mixture (viscosity 0.2052 St), 800 mL glycerol + 200 mL deionized water mixture (viscosity 0.690 St)). In the study, it was found that increasing the viscosity slowed down the process of cavitation bubble contraction and increased the minimum volume of the cavitation bubble before collapsing. In the most revealing comparison between presumably deionized water and an 80% solution (800 mL glycerin and 200 mL deionized water), the 80% solution had a 74% larger minimum bubble radius. It was also observed that as the viscosity of the solution increased, the velocity of the micro-jet produced during the first cycle of cavitation shrinkage of the bubbles decreased significantly. The authors were also able to draw another conclusion, stating that as the viscosity of the liquid increases, the compressive shock pressure on the surface of the wall decreases.

### 3.2. Oil Desulphurization

Sulfur is found in all grades of produced oil, and its content varies from 0.3% for low-sulfur oil to 3–8% for high-sulfur oil. The main forms of sulfur in oil and oil products are elemental sulfur; hydrogen sulfide; sulfides; disulfides; mercaptans; combinations of sulfur with oxygen, nitrogen, etc., which form complex compounds.

Sulfur is found in all grades of produced oil, and its content varies from 0.3% for low-sulfur oil to 3–8% for high-sulfur oil. The main forms of sulfur in oil and oil products are elemental sulfur; hydrogen sulfide; sulfides; disulfides; mercaptans; combinations of sulfur with oxygen, nitrogen, etc., which form complex compounds. In general, traditional methods of desulphurization of oil and petroleum products are associated either with the destruction and removal of sulfur compounds or with the selective extraction of sulfur-containing impurities from the oil. The content of any sulfur-containing impurities is unacceptable. Firstly, the use of sulfur-containing fuel may lead to additional pollution of the environment. Secondly, the sulfur content gives the fuel an unpleasant smell and causes corrosion of equipment. Thirdly, laws banning the use of high-sulfur fuels are becoming more common due to the exacerbation of the greenhouse effect, and the sulfur content causes poisoning of catalysts in the secondary treatment of petroleum products.

Experiments investigating the effect of acoustic cavitation on oil and petroleum products are the most frequently mentioned in the scientific literature. The effectiveness of acoustic cavitation is confirmed by many scientific papers [77,78]. In many papers [79–82] ultrasonic cavitation is used in conjunction with various oxidizing agents, which makes it difficult to carry out a thorough analysis of the effectiveness of acoustic cavitation directly on the desulfurization processes of raw materials. The effect of acoustic cavitation is associated with the creation at the implosion of cavitation bubbles in a microenvironment with extreme local conditions, allowing for the acceleration of the course of chemical reactions [83]. Earlier work carried out by researchers at BP Chemicals [84] also showed that the oxidation reaction can proceed without the use of ultrasound but at a much lower velocity.

In [79], the authors noted that acoustic cavitation was used only for increasing the efficiency of the oxidizer mixture (formic acid/acetic acid and 30% hydrogen peroxide). In [77] it is noted that the removal of sulfur decreases as the raw material is removed from the sound source due to the low amplitude of the ultrasonic wave.

A promising process for oil desulphurization has been hydrodynamic cavitation, widely studied by scientists in the industries of sterilization and wastewater disinfection. An increase in the overall intensity of cavitation can lead to the formation of reducing particles as a result of the thermal dissociation of hydrocarbon molecules, leading to the collapse of cavitation bubbles. This fact may lead to the formation of reducing compounds such as H and CO, which act as oxidizing agents (i.e., -OH and -HO2 are radical species, as opposed to oxidizing agents per se) [85]. Another important limiting factor is the excessive presence of hydroxyl radicals, which may lead to impairment of sulfur removal due to the absorption effect of hydroperoxyl radicals and other side reactions (i.e., oxidation of other organic compounds) as well as decarboxylation of formic acid.

Influence of the Initial Sulphur Concentration on the Degree of Removal

In the publication [86], there is one important aspect, often omitted by scientists, connected with the dependence of the initial concentration of sulfur on the efficiency of the desulphurization process of liquid hydrocarbon raw material. The treatment of liquid feed with a sulfur content of 300 ppm showed higher desulphurization results than the same treatment of feed with a sulfur content of 100 ppm. This result was explained by the authors with a simple and obvious hypothesis, according to which a higher content of sulfur compounds per unit volume of the liquid feedstock ensures a more frequent occurrence of compounds near the collapsing cavitation bubbles. Table 1 presents the results of experimental studies of the process of desulfurization of hydrocarbon products by cavitation, described in the scientific literature.

**Table 1.** Review of the effect of cavitation on the process of desulfurization of petroleum products described in the literature.

| Processed Product | Initial Sulfur Content | Type of Sulfur | Degree of Desulfurization | Processing Method |
|---|---|---|---|---|
| Hydrotreated Middle East Diesel [87] | 568.75 mcg/g | - | 93.3% | Ultrasonic device with 28 kHz ultrasound frequency, treatment time 15 min. US–Fenton reagent system with Fe+/$H_2O_2$ 0.05 mol/mol. |
| Synthetic fuel (n-octane number (Lobachemie, 98%), n-octanol (Lobachemie, 99%), toluene (Merck, >99%) and commercial diesel fuel (locally obtained)) [86] | n-Octanol—310 ppm; Toluene—290 ppm; Diesel—330 ppm. | - | n-Octanol—95%; Toluene—37%; Diesel—90%. | Diaphragm installation (one orifice, 3 mm). Differential pressure 5 bar. The content of the organic part relative to the water phase with sulfur—2.5%. |
| Hydrotreated feedstock Hydrogen content (%) 13.6 Carbon content (%) 86.4 Sulfur (initial, mg kg$^{-1}$) 3.6 Relative density 20 °C/4 (g/cm$^3$) 0.8362 Viscosity (cst, 20 °C) 10.0100 [83] | 3.6 mg kg$^{-1}$ | Dibenzothiophene (DBT, C1298 S, ≈98% purity, Merck, Darmstadt, Germany) | 95% | 20 kHz ultrasound machine. (9 min of treatment). Composition of the treated product: 15 mL acetic acid, 10 mL hydrogen peroxide, and 25 mL oil. The treated feedstock after irradiation was additionally extracted three times with polar solvent (MeCN, MeOH, or $H_2O$) using a glass separating funnel by manual shaking. |
| Diesel oil (fuel) (Sinclair Oil) [82] | 1. 0.7744 wt% (18 min irradiation); 2. 0.3011 wt% (10 min irradiation); 3. 0.1867 wt% (10 min irradiation). | - | 1. 0.0142 wt% (98.2%) 2. 0.0039 wt% (98.7%) 3. 0.0012 wt% (99.4%) | Ultrasound machine (VCX-600) (Sonics and Materials, Inc., Fleetwood, Pennsylvania, USA) 20 kHz. The polar solvent is acetonitrile. The temperature in the reactor is 75 ± 2 °C. Composition of the treated feedstock: Diesel fuel (oil); tetraoctylammonium bromide, phosphovol-tungstic acid, aqueous solution of $H_2O_2$. |

### 3.3. Increasing the Light Yield of Fractions

Many scientific publications [83,88,89] confirm the effectiveness of the effect of cavitation on the process of desulfurization of hydrocarbon raw materials.

In [90], results of the impact of intensive motion of ferromagnetic elements (Brownian motion) in the vortex layer activator on hydrocarbon feedstock are presented. The authors assume that under the influence of an external electromagnetic field, the ferromagnetic elements are remagnetized creating a magnetostriction effect that initiates acoustic cavitation. According to the statements of the authors, treatment by acoustic cavitation can increase the yield of gasoline fraction by 22.38%. However, it is worth noting that the authors did not evaluate the cavitation created. The effect of magnetostriction in devices of such design is rather small, which is also confirmed by the results of other studies. According to studies [91,92], the energy used for the share of acoustic vibrations does not exceed 2%. Thus, in the presented work, the effect of increasing the yield of "light" fractions is probably not acoustic cavitation, but intensive collisions of ferromagnetic elements with each other and with the walls of the working chamber. According to some studies [92], given the size of a single ferromagnetic element and the speed of movement in the volume of the working chamber, the substance between the two elements is put under a pressure of 300 MPa at the moment of their collision.

Despite the previous work, the basic part of experimental research is occupied with studying the influence of cavitation in Venturi tubes on the hydrocarbon composition of oil products. In [93], the implementation of 10 cycles of petroleum product passages through the tube's neck was noted as a decrease in boiling point and density during the processing of processed raw material. After five cycles of processing, the boiling point dropped from 280 °C to 250 °C. Considering the authors' statements that hydrodynamic cavitation changes the fractional composition of the feedstock, the work does not present research directly on the fractional composition.

In [94], a method of enrichment of a wide range of liquid hydrocarbons was patented. Application of this method based on acoustic cavitation made it possible to increase the yield of lighter fractions by 14–60%, depending on the type of treated feedstock.

### 3.4. Influence of Liquid Parameters on Cavitation Rate

Cavitation is a complex phenomenon and the probability of its occurrence in a fluid medium depends on many parameters, including both the type of device, its dimensions, the materials used (surface roughness [95]) and dimensions, as well as the characteristics of the feedstock, including rheological properties of the fluids, presence of mechanical impurities, gas inclusions, and other liquid inhomogeneities or cavitation cores. Cavitation nuclei are impurities, such as undissolved gases and solids, on which microscopic gas bubbles form [96]. Cavitation formation can be homogeneous or heterogeneous. In a liquid that is free of mechanical impurities and gas inclusions, the formation of cavitation bubbles is difficult and is predominantly homogeneous, and is created by the separation of the liquid molecules. [96]. According to the results of [97] cavitation in pure deionized water, the threshold for the formation of cavitation bubbles at room temperature is −60 MPa. However, in water with gas inclusions and mechanical impurities, the threshold of cavitation bubble formation according to the results of [98] is 0.1 Mpa. In turn, bubble dynamics are also related to many different factors, including surface tension and viscosity.

Temperature and viscosity also have a significant influence on the creation of cavitation. Cavitation treatment of a multi-component fluid such as oil can affect both temperature and viscosity. Treating oil with devices with moving internal elements can transfer heat to the treated liquid, raising its temperature and thus affecting viscosity. However, the hydrogen donor has a greater effect on the viscosity parameter [99], which can either be added externally according to the process or directly generated from the feedstock itself. As a result, the established initial parameters of the cavitation generator in the treatment process may not be sufficient to maintain stable cavitation in the treated feedstock.

In [41], a detailed study of parameters affecting the intensity of generated cavitation was carried out on the example of liquid treatment in a Venturi tube. In the study, the authors noted that the probability of cavitation bubble formation is affected by several factors, including the geometry of the tube constriction, medium temperature, liquid density, and the size of cavitation nuclei. Under the preservation of hydrodynamic conditions and insignificant change in the geometry of tube constriction, a significant change in cavitation intensity was observed. The effect of thermal lag of cavitation has been observed [100], according to which the vapor density increases along with the rise of feedstock temperature. As the cavitation bubble grows through the vapor, a decrease in liquid temperature and a local drop-in evaporation pressure are observed. When the bubble collapses, there is a sudden recovery of temperature, which eventually exceeds the initial temperature. Lower pressure is then initiated by the formation of the next cavitation bubble, which enlarges into a smaller bubble [41]. Similarly, in [41] the effect of the number of cavitation nuclei on the cavitation intensity has been investigated. At identical parameters of dimensionless cavitation number, flow velocity, and visually identical cavitation in treated liquid containing a smaller number of cavitation nuclei, significantly lower pressure fluctuations were observed, indicating a lower aggressiveness of cavitation.

Based on the above, it is evident that the use of dimensionless cavitation number as a single parameter to estimate the probability of occurrence and intensity of the cavitation generated is not appropriate.

## 4. Discussion

### 4.1. Aspect of Hydrocarbon Fractional Composition

In the works considered, the authors of the studies used hydrocarbon liquids from different fields. Crude oil is a complex multi-dispersed liquid substance. In each field, the fractional composition will have certain peculiarities, oil will have its rheological properties, different quantity of mechanical impurities, and gas inclusions, which can influence not only the cavitation threshold but also the cavitation rate and accordingly the treatment efficiency.

As the result of each study, the effect of cavitation on the new unique liquid medium with its unique fractional composition, including the content of paraffin, asphaltenes, hydrocarbon chains, and other compounds, as well as mechanical impurities and gas inclusions having significant influence on the overall rheological means, was investigated. It is practically impossible to confirm the results of studies presented in scientific works on other grades of oil products because of significant differences in the composition and properties of oil product grades from different fields. The number of impurities and gas inclusions will have a direct influence on the cavitation threshold and its intensity. Different fractional compositions of each kind of oil product will react differently to cavitation influence. As a result, both the parameters of cavitation action and the overall efficiency of the treatment process will differ if the oil grade is changed, and the initial parameters of the cavitation action system are maintained.

Until an unambiguous relationship is found between the rheological characteristics of the liquid feedstock, the number of gas inclusions, mechanical impurities, and the intensity of cavitation, it will be difficult to compare the results of experimental studies.

### 4.2. Cavitation Aspect

The creation of stable cavitation, whether acoustic or hydrodynamic, is not a simple process, depending on many primary parameters, including velocity, pressure, or viscosity, but also on secondary parameters such as the number of gas inclusions and mechanical impurities. The threshold for cavitation creation and its intensity are also influenced by external system components that do not directly create cavitation. For example, in many studies, it was noted that a pump standing in front of the Venturi tube in the system creates turbulent flow, which is then fed into the tube, simplifying the creation of hydrodynamic cavitation. When describing experimental studies, scientists still rely on

the dimensionless cavitation number. The dimensionless cavitation number describes the conditions of cavitation through inlet pressures or flow velocity. Based on the previously presented parameters, it can be concluded that cavitation depends on many interdependent characteristics of the cavitation system and the properties of the raw material to be treated. Thus, a single parameter cannot fully describe the results and is misleading.

*4.3. Efficiency Aspect of Raw Material Treatment*

Undoubtedly, any cavitation-generating installation is built for maximum efficiency. Some researchers try to measure the rate of cavitation through the efficiency of the treatment process, not taking into consideration that besides cavitation itself, the treated product can be also influenced in one way or another by accompanying energetic effects of the basic technological process, such as mechanical effects of colliding elements or additional influence of chemicals and chemical elements that strengthen or weaken the quality of the treatment.

*4.4. Recommendations for Further Studies of the Amount of Cavitation on the Fractional Composition of Oil*

The main drawback of most of the studies studied in this scientific review is the lack of a precise description of both the geometry of the cavitation generator and the characteristics of the raw material to be treated and the parameters of the treatment process. Considering that cavitation depends on many parameters, the authors, when presenting the results of experimental studies, should indicate the exact description of the geometry of the applied cavitation generator, parameters of accompanying equipment (compressor), characteristics of initial oil products, including fractional composition, temperature, and rheological properties. It is also necessary to specify treatment process parameters, including treatment duration, cavitation intensity, and dimensionless cavitation number to study this parameter, including the exact location of measuring the dimensionless cavitation number.

The presented description of research will allow scientists not only to reproduce earlier experimental studies but also to accelerate the development of this scientific field.

In many of the works presented, aimed at studying the treatment of liquid petroleum products, the authors used a combination of the effects of cavitation and additional chemical reagents. Taking into consideration many different possible reagents for application and their different influence on the initial product, these results do not allow us to define the exact degree of influence and the role of cavitation. To make a complete evaluation of the method's effectiveness, the investigators should present, in addition to the joint application of cavitation and a chemical, the possibility of implementing cavitation separately on the petroleum product under study as well as the reagent. If possible, research should be accompanied by the study of separate effects of cavitation and chemicals on individual fractions of petroleum products.

Notwithstanding a great number of successful results of experimental investigations of the cavitation effect on various liquid feedstock, including oil and liquid petroleum products, at the moment there is no wide application of given technology in the industry. The main factor that prevents the introduction of cavitation influence technology into the industry is the difficulty of reproducibility of experimental results after changing the parameters of both the generator geometry and the raw material. Up to date, the intensity of cavitation in most cases is determined only by experimental methods, which is inadmissible for big industrial enterprises in the development and production of technological lines for the processing of oil and liquid oil products of industrial scale that require big investments at high probability of negative results. Further investigations should be based first of all on searching for a unified methodology of estimation of cavitation intensity that will predict changes in the intensity of created cavitation at any change in technological process parameters.

In general, investigations of the energy effects of cavitation and their effects on various physical/chemical, etc., properties of the processed raw materials, for effective use of

cavitation generators seem to be interdisciplinary research and should be carried out by joint efforts of scientists representing such fields of science as chemistry, engineering, physics, and mathematics.

*4.5. Recommendations for Finding New Expressions of the Cavitation Number*

On the basis of the review, it was noted that the main problem that needs to be solved in the near future is the lack of a unified methodology for the application of cavitation in the treatment of liquid raw materials. As a result, each individual scientific team in the framework of their studies is guided by their own ideas about both the mechanism of cavitation effect on the studied fractional composition and directly about the mechanisms of cavitation formation in the liquid medium and the conditions affecting its intensity. The lack of reliable, complete, and accurate description of the technological process of liquid petroleum product treatment only hinders further development of this field of science as every single result of the experimental investigation is not related to others, practically cannot be reproduced and in absence of information about the technological process, it has only informative character and does not allow us to apply it in further investigations.

Moreover, many authors, when presenting the results of their scientific work in the form of publications, omit incredibly important data on the parameters of processed liquid raw materials. The result in many scientific papers is presented by experimental data on the treatment of this or that raw material with an indication of percentage ratio before and after, and in some cases with an indication of final fractional composition. As a result, any attempts to independently reproduce the cavitation treatment of liquid raw materials cannot be successful and can neither confirm nor refute the previously presented results.

Thus, at present, it is quite difficult to evaluate and compare the results available in a wide range of studies, because these considerations are not always taken into account by the authors. Given the uniqueness of each installation, it is difficult to conclude the degree of influence of such parameters as the velocity of the fluid in the working chamber, the de-construction of the working chamber, and the geometric parameters of the solid elements, whether it is the diffuser against which the fluid moves, or the secondary discrete element itself, which directly moves against the fluid. To create the possibility of comparing results, a uniform methodology for evaluating cavitation efficiency is needed. In a number of studies, the cavitation number calculated by standard methods is used in evaluating the intensity and threshold of cavitation creation. However, it cannot reliably describe cavitation in the liquid under study. In existing expressions describing the dimensionless cavitation number, only vapor density and pressure describe the characteristics of the liquid to be treated. The viscosity and surface tension of the liquid also influence the creation threshold and the intensity of cavitation but are not directly related to the vapor pressure or density used in the calculation of the cavitation number.

The influence of individual parameters in controlled laboratory studies is presented in many scientific papers. The influence of surface tension on the cavitation bubble dynamics was also evaluated in [101]. The results of this study showed that lower surface tension increases the deformation of the gas–liquid interface, which leads to a more concentrated microstructure. The degree to which the temperature of the treated liquid feedstock affects cavitation has been evaluated in [102–105]. Cavitation nuclei, including solid insoluble particles [106,107] and gas impurities [108] also have a high influence on the intensity and threshold of cavitation creation. Separately, it should be noted that cavitation is also influenced by the geometric parameters of the cavitation generator design.

Based on the theoretical studies, it becomes evident that finding a single expression for calculating the dimensionless cavitation number, taking into account the specifics of a large number of cavitation generators, is impossible. In real conditions at the processing of multicomponent liquids, the complexity of fractional composition has unpredictable mutual influences both on each other and on the intensity of cavitation. The lack of information about the mutual influence of different parameters on each other and on cavitation intensity does not allow us to predict the cavitation treatment of natural systems.

However, the complexity of cavitation also lies in the variability of the parameters of the treated raw material during the implementation of the proposed technological process. The collapse of cavitation bubbles, as mentioned above, leads to the release of energy and a local increase in temperature and pressure. The process of collapsing cavitation bubbles not only increases the temperature of raw materials and, accordingly, changes the viscosity of the liquid, but also changes the fractional composition of the liquid medium. In addition to increasing the temperature, cavitation treatment also affects the very fractional composition of the feedstock, including, by the example of crude oil, reducing the content of asphaltenes and paraffin, reducing the viscosity. It is the high content of asphaltene, paraffin, and alkane molecules that leads to an increase in oil density and viscosity and leads to changes in colloidal structure and rheological characteristics. The cavitation effect also dissipates the crystalline structure of paraffin in the crude oil, resulting in lower viscosity. In this case, both in the case of sulfur and paraffin and asphaltene molecules, the more molecules in the initial product, the faster the viscosity and density of the feedstock will decrease, because the higher the content of the corresponding molecules per unit volume of a substance, the more often the cavitation bubbles collapse in close proximity to the molecules. Additionally, the change in viscosity entails the need to make adjustments to the parameters of the units directly when implementing the technological process, which will also affect the processing time of the raw material. The accuracy of the prediction of viscosity and density decrease for multicomponent liquids can be studied only empirically because liquid hydrocarbons have a complex composition and are a mixture of more than 1000 different substances. At the same time, the mechanism of impact of cavitation on individual fractions and molecules of oil in most cases is still based on individual unproven hypotheses. Thus, it is impossible to unambiguously single out one or another parameter as the main one on which the treatment efficiency depends. Even with initially correctly calculated dimensionless cavitation number, and selected parameters of the technological system for the corresponding liquid substance, in the process of treatment conditions change, which will entail changes in the predicted intensity of cavitation. These conditions will require researchers to calculate not only the cavitation number at the beginning of work but also the dynamics of changes in the cavitation number of the implementation of the technological process, which will allow them to compensate for changes in the cavitation number when designing the processing system.

To date, the lack of a unified understanding of the mechanisms of mutual influence of the liquid and cavitation does not allow us to scale the technological process of oil treatment. Comparing the results of theoretical studies, it was concluded that the way to the introduction of cavitation in the oil refining industry is to find new expressions of the number of cavitation, allowing at the stage of plant design to evaluate the effectiveness of its application.

Creation of stable cavitation, whether acoustic or hydrodynamic, is not an easy process, depending on many primary parameters, including velocity, pressure, or viscosity, but also secondary parameters, such as the number of gas inclusions, mechanical impurities, the geometry of individual nodes of cavitation generator and others. The threshold of the creation of cavitation and its intensity is also influenced by external nodes of the system not used for the purposeful creation of cavitation. For example, in a number of studies, it is noted that a pump standing in the system before the Venturi tube creates turbulent flow, which is then fed into the tube, thereby facilitating the creation of hydrodynamic cavitation. Thus, further research should not rely on the evaluation of the influence of individual fluid parameters on cavitation or cavitation on individual raw material fractions, but on the gradual complication of experimental conditions by controlled sequential addition within the experiment of new variables affecting the threshold of creation and intensity of cavitation. Thus, the works should involve additional complicated experimental work to assess the mutual influence of individual fluid parameters and the geometry of cavitation generators on each other and analyze their overall impact on the change in the cavitation threshold. The controlled study will allow bringing the laboratory conditions closer to the

treatment of real fluids, which will simplify the derivation of the final new expressions of the cavitation number. Search for cavitation number should be carried out by non-linear regression analysis of the main parameters affecting the intensity and threshold of cavitation creation, allowing for correlating the beginning of cavitation with the main parameters of the technological process (fluid characteristics and geometry of the cavitation generator).

Such a nonlinear regression analysis for a specific generator and fluid geometry should rely on all the main parameters of the liquid feedstock obtained from the theoretical study, including the concentration of dissolved salts, presence of solid impurities (hydrophilic and hydrophobic), gas inclusions, surface tension, temperature, and viscosity. Regression analysis will make it possible to assess the degree of influence of individual raw material parameters and include them in the calculation of the dimensionless cavitation number. The results of such experimental studies will provide new empirical data to refine the dimensionless cavitation number. Selection of optimal parameters can be made on the basis of nonlinear regression analysis of most parameters of both processed liquid feedstock and geometry of this or that hydrodynamic cavitation generator:

$$\sigma\left(\Delta p,\ TDS,\ K_{\text{impurities}}, K_{\text{gases}},\ \eta,\ \rho,\ T,\ A\right) \tag{12}$$

where $\Delta p$—the difference between ambient pressure and saturated vapor pressure of the treated liquid, TDS—total amount of dissolved particles in water, mg/L, $K_{\text{impurities}}$—concentration of solid impurities (undissolved), mg/L, $K_{\text{gases}}$—dissolved gas concentration in liquid, mol/L, $\eta$—kinematic viscosity of the treated liquid, m$^2$/s, $\rho$—density of the treated liquid, kg/m$^3$, T—temperature of the treated liquid, °C, A—geometry factor of the hydrodynamic cavitation generator (for the Venturi tube, the nozzle convergence angle and the output diffuser divergence angle, for the device with a discrete secondary part, the convergence angle of the secondary discrete part element).

## 5. Conclusions

According to the presented analytical review of the scientific and technical literature, the results often do not correspond to each other. The main problem to be solved in the near future is the lack of a unified methodology for the application of cavitation in the treatment of liquid raw materials. As a result, each separate scientific group within the framework of its investigations is guided by its own ideas both about mechanisms of cavitation effect on investigated fractional composition and directly about mechanisms of cavitation formation in liquid medium and conditions that affect its intensity. The lack of a really full and accurate description of the technological process of liquid petroleum products processing only hinders further development of this field of science as every single result of the experimental investigation is not connected with others, practically cannot be reproduced repeatedly, and in absence of information on technological process has only an informative character that does not allow us to apply it in further investigations.

Comparing the results of theoretical studies, it was concluded that further research should not be based on the evaluation of the effect of individual parameters of the liquid on cavitation and not on studies of the effect of cavitation on individual fractions of hydrocarbons. The introduction of cavitation in the industry is impossible without finding a parameter that allows predicting and evaluating cavitation in the liquid under study, including liquid hydrocarbons. However, it is not possible to evaluate the effect of liquid properties directly in the studied product (natural system). To assess the influence of liquid properties it is necessary to conduct controlled studies of different mutual properties of raw materials on the intensity of cavitation with gradual complication of experimental conditions by controlled sequential addition of new variables (increasing viscosity, increasing the concentration of impurities, gases, or dissolved salts, etc.), that affect the threshold of cavitation creation. The controlled study will allow us to bring experimental studies closer to the treatment of real fluids and to identify the most significant liquid properties for the creation of cavitation, which will simplify the derivation of the final new expressions of the cavitation number. The cavitation number can be found by non-linear regression analysis of

the main parameters affecting the intensity and threshold of creation of cavitation, allowing for correlating the onset of cavitation with the main parameters of the technological process (fluid characteristics and geometry of the cavitation generator).

Nevertheless, given the decreasing reserves of "light" oil and the growing share of "heavy" oil in global oil production, scientists are faced with the task of finding new ways of oil refining. The high prevalence of this scientific trend, a large number of publications and patents, confirm the prospects of application of cavitation in various industries..

**Author Contributions:** Conceptualization, D.K.; methodology, D.K.; project administration, D.K. and M.M.; validation, D.K.; formal analysis, D.K.; investigation, D.K.; writing—original draft preparation, D.K.; writing—review and editing, D.K., M.M., A.Y. and A.L.; visualization, A.Y. and A.L.; supervision, D.K.; funding acquisition, D.K. All authors have read and agreed to the published version of the manuscript.

**Funding:** The study was supported by grant No. 22-79-00226 from the Russian Science Foundation, https://rscf.ru/project/22-79-00226/ (accessed on 30 May 2023).

**Data Availability Statement:** The data can be shared upon request.

**Conflicts of Interest:** The authors declare no conflict of interest.

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
