# Peer review of "Current State of Research on the Mechanism of Cavitation Effects in the Treatment of Liquid Petroleum Products—Review and Proposals for Further Research"

_fluids, doi:10.3390/fluids8060172_

Round 1

Reviewer 1 Report

In this work authors presents a review of scientific research aimed at studying the effects of cavitation on liquid petroleum products and oil residues. Before this work can be accepted some significant revisions must be made:
1. Introduction does not reflect the novelty of this work. There are large numbers of publications on the study of cavitation effects in the treatment. It s not clear why this work is different.

2. The titles of section 1.1 and 1.2 are repeated.

3. Significant mistakes are shown in Section 2. Materials and Methods. Please take this work seriously.

4. All the figures need to be redrawn by yourself. Please do not screenshot and paste directly.

5. Table 1 needs to be re-format.

6. The abstract needs to be extended.

7. The introduction should be extended and improved. The authors should better describe the previous works and underline the novelty of their study
Some suggestions to extend the literature review are given in the following

- ACS omega 6.47 (2021): 31411-31420. https://doi.org/10.1021/acsomega.1c05858

- Int. J. Heat Mass Transfer 170 (2021): 120970, https://doi.org/10.1016/j.ijheatmasstransfer.2021.120970;

- Ultrasonics Sonochemistry 70 (2021): 105337. https://doi.org/10.1016/j.ultsonch.2020.105337

- Energy 254 (2022): 124426, https://doi.org/10.1016/j.energy.2022.124426;

- Engineering (2022). https://doi.org/10.1016/j.eng.2022.04.027

- Ultrasonics Sonochemistry 86 (2022): 106035, https://doi.org/10.1016/j.ultsonch.2022.106035;

8. Conclusion is superficial : no any values and no any % are given.

9. The author states that “As the result each separate scientific team in the framework of its investigations is guided by its own ideas about both mechanism of cavitation effect on investigated fractional composition and directly about mechanisms of cavitation for-mation in liquid medium and conditions that influence on its intensity.” Then what is your solution?

The language requires major revision. The paper does not read smoothly (many typos) and it is redundant. I suggest the authors consult a native English speaker. 

Author Response

Point 1

The structure of the publication has been changed, including a revised and expanded introduction.

Point 2

The section titles have been edited.

Point 3

Thanks for the comment. The Materials and Methods section is rewritten.

Point 4

All the figures were redrawn.

Point 5

Since the information presented in the table did not provide any additional information, it was deleted.

Point 6

The abstract has been expanded to reflect changes in the publication.

Point 7

The introduction has been modified and expanded to reflect changes in the publication.

Point 8

The conclusion of the publication is expanded.

Point 9

The publication, with the changes in the article, has expanded the conclusion.

Reviewer 2 Report

I do think this review paper is well structured and has the clean logic & necessary depth to be suitable for publication. A few points for improvement:

1) Abstract is not informative enough and should be rewritten more scientifically. The current way it is written now does not address the importance and significance of the work, nor the overall knowledge that was produced.

2) Page 3 Line nos. 105-114: Please complement and enrich current statement with more related literatures: “Study on micro remaining oil distribution of polymer flooding in Class-II B oil layer of Daqing Oilfield(2022). Energy, 254” and “Modeling of microflow during viscoelastic polymer flooding in heterogenous reservoirs of Daqing Oilfield(2022). J. Pet. Sci. Eng. 210, 110091”.

3) Section 2: Materials and Method. The section is confusing. What is your main objective in this section? Please state clearly.

4) Page 5 Line nos. 214-215:, recent study on defoaming with ultrasonic standing wave can also be read and referred, "Modeling of kinetic characteristics of alkaline‑surfactant-polymer-strengthened foams decay under ultrasonic standing wave (2022). Petroleum Science, https://doi.org/10.1016/j.petsci.2022.04.012”.

I would recommend accepting this paper with minor revision.

Authors should review the text amending all typos from the text.

Reviewer 3 Report

The submission is well-written and the standard of English is generally very good. Having said that, if deemed publishable, there are aspects that would need to be tidied up.

However, in my view, I believe that the authors should reconsider the objectives of the submission, because the messages given seem to be largely based on the wide range of different cavitation devices, or quantification of cavitation, rather than the results of treating petroleum products, as the title implies. I would agree that this will lead to difficulties in comparing different studies, which is perhaps the main conclusion of the present work.

Having said that, I would also like to point out that the paper by Stepeleva and Minakov (cited as ref [13]) deals in a more useful way with the potential application of cavitation in the petroleum industry, which almost negates the need for the present review. The present submission is weak on physical/chemical mechanisms. Aspects of scale-up should also be included if considering commercial processes.

Should the authors wish to consider changing the real direction of their study, I would suggest that they could produce a more succinct analysis of the differences in cavitation apparatuses and methodology, towards a more unified approach that would be applicable to a range of feedstocks.

As it stands, the present study lacks sufficient detail on the chemical (and physical) reactions and processes to which cavitation is targeted. This is one aspect where the aforementioned ref [13] has an advantage. The authors have chosen a field that is till in its infancy, in many ways, which is the reason for many different types of approach being taken.

The authors should present an indication of the amount of acoustic/cavitation energy (per unit volume) required to generate the chemical energy required to produce the results required for industrial processes.

There are a few specific points that ought to be made regarding the paper itself:

Lines 160-174: Section 2 - This should be removed, as it is a reproduction of the publisher's information.

Line 359: Section 3.3 - This should be considered and re-written for improved clarification.

Line 374: delete the authors' initials. In fact, most reference numbers have not been properly incorporated into the text, and being associated with the authors or a specific statement or results. There are some exceptions, but not many.

Line 388: delete "in inorganic substances"

Line 389: delete "in organic substances"

Line 394: delete "as a reagent"

Line 398: delete "as a device"

Lines 413-414: although paraffinic oils do contain resins

Lines 563-567: include viscosity units

Line 570: presumably deionised water

Table 1: No reference is made in the text to the content of this table. Requires discussion

Line 599: BP Chemicals

Line 610: -OH and -HO2 are radical species, as opposed to oxidising agents per se

Table 2:  Again, no reference made to the content of this table in the text. Requires discussion

Lines 628-630: not a complete sentence

Lines 659-662: re-write to improve

Line 669: ... postulates have stood ...

Line 702: re-write to improve

Lines 737-739: not a proper sentence

Lines 744 and 800: replace disperse with component?

Line 765: delete "content"

Line 777: use "individual", rather than single copy?

Lines 805-809: re-write this paragraph to clarify

Lines 817-819: is this not likely to be an almost impossible task? Needs further discussion

Line 824: thresholds

Line 859: ... not allow one to define ...

Line 863-865: this has been done previously with crude oil asphaltenes

Lines 884-885: "presented" used twice

English needs some attention, but is generally good.

Round 2

Reviewer 1 Report

All the comments are addressed.

Minor editing of the English language is recommended.

Reviewer 3 Report

There are still a few issues that I hope will be picked up on editing prior to publication. For example, the use of commas for decimal points is still evident in the manuscript - these have been changes in some places, but not all. In Table 1 there is some Russian script.

Overall, this is fine and should serve as a useful reference for previous studies on cavitation in relation to petroleum substrates.